# Microneedle-Integrated Sensors for Extraction of Skin Interstitial Fluid and Metabolic Analysis

**DOI:** 10.3390/ijms24129882

**Published:** 2023-06-08

**Authors:** Jie Yang, Ruiyu Luo, Lei Yang, Xiaocheng Wang, Yong Huang

**Affiliations:** 1State Key Laboratory of Targeting Oncology, National Center for International Research of Bio-Targeting Theranostics, Guangxi Key Laboratory of Bio-Targeting Theranostics, Collaborative Innovation Center for Targeting Tumor Diagnosis and Therapy, Guangxi Medical University, Nanning 530021, China; yangj5187@163.com (J.Y.); ruiyu0401@126.com (R.L.); 2Oujiang Laboratory (Zhejiang Lab for Regenerative Medicine, Vision and Brain Health), Wenzhou Institute, University of Chinese Academy of Sciences, Wenzhou 325001, China; yangleigeili@163.com

**Keywords:** microneedle, biosensor, ISF sampling, transdermal detection, diagnostic technique

## Abstract

Skin interstitial fluid (ISF) has emerged as a fungible biofluid sample for blood serum and plasma for disease diagnosis and therapy. The sampling of skin ISF is highly desirable considering its easy accessibility, no damage to blood vessels, and reduced risk of infection. Particularly, skin ISF can be sampled using microneedle (MN)-based platforms in the skin tissues, which exhibit multiple advantages including minimal invasion of the skin tissues, less pain, ease of carrying, capacity for continuous monitoring, etc. In this review, we focus on the current development of microneedle-integrated transdermal sensors for collecting ISF and detecting specific disease biomarkers. Firstly, we discussed and classified microneedles according to their structural design, including solid MNs, hollow MNs, porous MNs, and coated MNs. Subsequently, we elaborate on the construction of MN-integrated sensors for metabolic analysis with highlights on the electrochemical, fluorescent, chemical chromogenic, immunodiagnostic, and molecular diagnostic MN-integrated sensors. Finally, we discuss the current challenges and future direction for developing MN-based platforms for ISF extraction and sensing applications.

## 1. Introduction

Identification and detection of biomarkers in body fluids have become increasingly important in advanced medical diagnosis and disease therapy [1,2,3,4]. Biofluid sampling is the first and essential step for biomarker-associated in vivo metabolic analysis. Up to now, the blood is still the best common source of biomarkers, which can provide important information for diagnosing disease, judging prognosis, and monitoring physiological metabolism [2,5,6,7,8]. Unfortunately, the blood collection procedure often involves or triggers tissue damage, potential infection, and unavoidable pain [9,10]. Other biofluids (e.g., urine, saliva, sweat, and tears) are usually limited by their poor correlation with diseases due to the significantly variable levels of biomarkers [11,12]. Nowadays, skin interstitial fluid (ISF), which originates from capillary filtration through blood to the intercellular substance of the dermis, has emerged as an alternative biofluid source to blood serum and plasma for medical diagnosis and disease therapy [13]. The sampling of skin ISF is highly desirable mainly because of its easy accessibility, no damage to blood vessels, and reduced risk of infection [14,15]. At present, increasing biofluid sampling methods have been explored, such as the wick method [16,17,18,19,20], suction blisters [21,22], microdialysis and ultrafiltration [23,24], needle-based extraction [25], reverse iontophoresis (electroosmosis) [26,27,28], capture needles [18,29], and so on. Among them, needle-based extraction has been recognized as the most common and reliable method in clinical and translational settings [30,31]. 

Microneedles (MNs), well-known as micron-scale needles, can penetrate the outer skin barrier (i.e., stratum corneum) and collect cutaneous ISF in a minimally invasive way [32,33,34]. The MNs patches have been widely explored in transdermal drug delivery for over two decades, and they have been well-accepted by patients in clinical trials and other studies owing to their safe, effective, and painless characteristics [35,36]. More recently, people have paid more attention to the MN-based flatforms for skin ISF sampling [37]. Based on the different microstructures of MNs for ISF collection and extraction, current MNs can be divided into solid MNs, hollow MNs, porous MNs, and coated MNs [38]. Compared with traditional ISF extraction methods, MNs show significant advantages in practice [39]. Typically, no special equipment is required for MN-based platforms. Meanwhile, the reduction of skin puncture pain and improvement of bioavailability can be further achieved via optimization of the MN components and structures [40,41]. More importantly, portable sensing devices can be precisely integrated with MNs [12,42]. Thus, MN-based wearable sensors can realize specific biomarker capture, recognition, in situ detection, and continuous monitoring in the skin ISF [43]. These advantages make the MN-integrated sensors powerful platforms for skin ISF extraction and sensing applications. Although some examples of microneedle sensors were developed for other biological samples instead of ISF in this review, the design concepts and working principles behind them could provide constructive inspiration for the development of more advanced MN-integrated sensors for the extraction and metabolite detection of ISF.

In this review, we will systematically describe the construction and related operating principles of MN’s integrated sensors for ISF extraction and metabolic analysis (Figure 1). We begin with the structural design of MNs for collecting and extracting ISF. Then we elaborate on the construction of MN-integrated sensors for metabolic analysis with other detection methods. Finally, we discuss the potential for future development of MN-based platforms for ISF extraction and sensing. We hope this review can lay the foundation for the research of advanced MN-integrated sensors for the analysis of biomarkers in skin ISF and facilitate their translation for future clinical applications.

## 2. MN Structure Designs

In recent years, MNs have attracted more and more researchers’ interest due to their potential in minimally invasive diagnosis and have gradually been applied to extracting ISF from skin tissues (Table 1). MNs consist of a series of micron-scale needles with a shallow penetration depth, providing a minimally invasive method to puncture the skin in a painless or slightly painful manner [44,45]. It is worth mentioning that the microstructure of MNs tips plays a decisive role in ISF extraction. Based on the MN structures, current MNs used for ISF extraction can be classified into solid MNs, hollow MNs, porous MNs, and coated MNs. 

### 2.1. Solid MNs

Solid MNs represent a class of MNs without gaps inside their tips, and most of them are made from macromolecular hydrogels and widely utilized for transdermal drug delivery [42]. Hydrogel-solid MNs for ISF extraction are hard in the dry state, so their needle tips can easily penetrate the skin barriers [11]. When retained in the skin tissues, the tips can swell and actively absorb ISF containing biological small molecules through gradient diffusion such as a sponge, realizing in situ ISF extraction from local tissues (Figure 2a) [47]. The absorbed biological molecules will be further separated from hydrogel MNs by centrifugation or other separation methods, and specific biomarkers can be detected in combination with existing analytical methods, such as high-performance liquid chromatography (HPLC), enzyme-linked immunosorbent assay (ELISA), etc. [3,47]. 

Normally, solid MNs are used to collect and extract ISF for in vitro detection of cholesterol, glucose, nucleic acid, protein, drugs (such as vancomycin), and so on [55]. In one example, SeungHyun Park et al. exploited solid methacrylated hyaluronic acid (MeHA) MNs for ISF extraction and dopamine detection (Figure 2b) [37]. Benefiting from the MeHA hydrogels’ supreme water affinity, they achieved a high in vitro swelling ratio (600% within 30 s) (Figure 2c). In another example, Qiqi Yang et al. also used MeHA MNs with high MeHA concentrations as the hydrogel backbone to shorten the sampling time and improve the extraction efficiency [56]. Such solid MNs can extract 0.97 ± 0.2 mg miRNA from adequate ISF within 5 min. In addition, the extraction of target substances could be accelerated by increasing the osmotic pressure within the ISF. For example, maltose is used as an osmotic agent to enhance the IFS osmotic pressure and thereby facilitate the ISF diffusion from the skin to the hydrogel [47]. The MeHA MN containing 100 maltose-containing tips could accelerate ISF extraction from pig skin (7.90 μL in 3 min). By contrast, the control group (MeHA MNs without maltose) took at least 10 min to achieve similar effects. Therefore, the swellable hydrogel MNs can be applied to biomolecule sensors and detect biomarkers of ISF in a short time, providing a powerful system for skin ISF extraction [46].

In fact, most solid MNs are made up of photopolymerized hydrogels, which are easily prepared by mold methods. As shown in Figure 2d, the pre-gel solution was added into a PDMS mold, gradually filled all cavities via vacuum or centrifugation, and subsequently crosslinked under UV irradiation. After drying, a rigid, solid MN with a flat surface structure was obtained. In addition to MeHA [37,47,56], macromolecular hydrogels such as gelatin methacryloyl (GelMA) [48], poly (ethylene glycol) diacrylate (PEGDA) [57,58,59], and methacrylated hyaluronic acid (MAHA) [60,61,62] have been used to fabricate solid MNs. Solid MNs have an inherent structural advantage, which provides them with increased mechanical strength and thus facilitates insertion into skin with a reduced risk of fracture [52]. Based on the hydrogel swelling behaviors and adjustable osmotic pressure of ISF, solid MNs are feasible to detect small molecular substances (e.g., drugs and glucose), but they hardly capture insoluble macromolecular substances (such as cells). 

### 2.2. Hollow MNs

Hollow MNs are a kind of MN with a hollow channel inside the needle tips. Biomarkers in ISF can be collected by flowing through a network of microcapillaries in a single capillary using hollow MNs [12] (Figure 3a). Unlike the capillary blood collection currently used in the clinic, which requires puncturing the skin with a hollow steel needle. The hollow MNs can usually be used to extract sufficient skin ISF with a painless method [50]. Compared with solid MNs with better swelling performance, hollow MNs usually choose materials with non-swelling properties and high mechanical strength, such as silicon (Si) and titanium (Ti) [12]. Most of them are precisely micro-machined via photolithography technology to endow hollow MNs with capillary-like microstructures [49,63].

In early studies, hollow MNs were fabricated by a series of procedures, including standard lithography, buffered hydrofluoric acid (HF) etching, and potassium hydroxide (KOH) etching (Figure 3b) [64]. The microstructure of hollow MNs obtained from multiple etching methods can be observed by SEM imaging (Figure 3c) [50], and these hollow MNs were used to detect glucose in ISF. However, the skin penetration efficiency will be greatly reduced if their tips are not sharp enough. Therefore, a new kind of MN was developed with sharp hollow stainless-steel tips, and a glass capillary was connected to the tail of the hollow MN’s tips (Figure 3d) [6]. After application for a lag time of 30–120 s, ISF flow was extracted into the capillary and continued for 10–15 min (Figure 3e). Although the device is not attributed to the metabolite detection in ISF, it presents a good example of how to integrate negative pressure devices with MNs to enhance MN extraction efficiency. The blood was still the gold-standard sample medium. If we could extract blood directly from capillaries using MNs, which would thereby facilitate the detection of abundant useful biological information for the diagnosis of various diseases. To enhance the extraction efficiency of hollow MNs, a negative pressure device can be added to the MNs back to accelerate the extraction rate. As an example, Cheng Guo Li et al. introduced a blood extraction device that was fabricated by connecting hollow MNs and a pre-vacuum PDMS actuator (Figure 3f) [65]. Because of the capillary action of hollow MNs and the negative pressure force from the actuator, it successfully extracted sufficient blood (31.3 ± 2.0 μL) from a rabbit (Figure 3g), yielding sufficient volume for use in a microanalysis system. In contrast, a large quantity of body fluids was hard to extract without a pre-vacuum system. As a result, the pre-vacuum activation system is compact and simple to operate and can meet the challenging requirements of single-use applications without any external power supply.

Recently, in order to improve the fabrication efficiency and reduce the process steps, Anika Trautmann et al. reported a novel hybrid method that combined a femtosecond direct laser programmed into a MN patch with a femtosecond laser-generated microfluidic channel [66]. The ultra-short pulsed laser system can build 3D microchannels directly inside the polymethyl methacrylate (PMMA) bulk material, which have adjustable cross-section areas and permit the MNs to attach to the microfluidic system. Therefore, hollow MN arrays were prepared using a laser system designed for two-photon polymerization (Figure 3h). Such a design overcomes complex fabrication methods, and the manufacture of the MNs is processed by programming. Unfortunately, the high risk of single channel clogging is considered a major problem in the ISF extraction of hollow MNs. Later, the researchers moved the hollow channel toward the edge to preserve both the sharp tip and the hollow microchannel (Figure 3i) [67]. The micropillars incorporating holes could be sharpened using a mixed HNO_3_ and HF solution, and the hollow channel was retained at the edge. Similarly, Wu et al. made hollow MNs with an oblique channel such as a hypodermic syringe (Figure 3j) [68]. The MN tips were quite sharp to effectively pierce the dermis, while the hollow channel could efficiently collect ISF. In another case, Smith et al. eased the clogging problem by moving the channel to the edge of the Si MN and forming a snakelike hollow MN [69].

### 2.3. Porous MNs

The tip of porous MNs usually contains a high density of capillary network structures [70,71,72], and its pore size ranges from tens of nanometers to several microns [8,73]. Similar to the extraction principle of hollow MNs, ISF can be extracted into their tips and even their substrate by capillary continuous extraction (Figure 4a) [12]. It is worth mentioning that even though one channel is blocked in porous MNs, other channels can still work to extract ISF.

Porous MNs could be fabricated by the simplest yet most effective mold method without involving high temperatures, high pressure, or nuclear irradiation, which often used some highly porous and interconnected polymers (Figure 4b) [74]. These porous MNs displayed great capacities in collecting and extracting ISF applications due to their multi-porous structures and high porosity. Some common polymers include cellulose acetate (CA), polysulfone (PSF), polylactic acid (PLA), etc. [12,55]. For example, Pei Liu et al. reported porous CA MNs with highly efficient extraction of ISF and glucose level detection [74]. The porous and interconnected structures enabled the CA MNs to extract effective ISF for offline glucose detection in the dermis. Specifically, the porosity rate of the CA MNs could be well controlled at 90–40% by tuning initial CA concentrations to 10–35 wt%. In another case, Leilei Bao et al. fabricated biodegradable porous PLA MNs to extract SARS-CoV-2 IgM/IgG-antibody from human skin [75], where the detection limit of antibodies was as low as 3 ng/mL (IgM) and 7 ng/mL (IgG). In general, these porous polymer MNs have a relatively simple preparation process without the need for special techniques for pore formation. Such spontaneously formed porous MNs have relatively high porosity but poor mechanical strength.

In addition, it is also common for MNs to create pores by introducing pore-forming agents, and in this case, the materials usually possess a higher mechanical strength to stabilize the pore size of MNs [76,77]. For example, Jian Zhang et al. prepared porous MN by the salt leaching method [53]. The desirable pores inside the MNs were fabricated by mixing acrylic resin and salt, followed by etching or washing out the salt with water. The pore size was based on the sodium chloride particles with a size less than 40 μm, which was small enough for efficiently collecting glucose ISF and provided a good interconnected porous structure inside the MNs (Figure 4c). Moreover, the porous structure enabled effective extraction of ISF from the epidermis and dermis with a significantly reduced lag time, which greatly facilitated in vivo glucose monitoring. Similarly, Shinya Kusama et al. prepared the porous MNs from a porogen stock solution by dissolving polyethylene glycol in 2-methoxyethanol [78]. For another example, porous MNs were manufactured by loading glass microspheres with different amounts and sizes of pore-forming reagents [79]. In this case, the porous structure was obtained by removing the glass microspheres using an HF solution (Figure 4d). It is noted that the pore size and interconnection structure of the porous MNs could be precisely controlled by using different pore-forming agents with variable sizes.

In addition, metal materials such as titanium, nickel, and stainless steel have also been employed to prepare the porous MNs for collecting biofluids. Considering their mechanical properties and electrical conductivity, porous metal MNs are commonly fabricated by electroplating, electrode wire cutting, and laser cutting techniques [80]. For example, Ellen M. Cahill et al. prepared porous 316 stainless steel MNs by electropolishing the steel after calcination [52]. The manufacturing process could be optimized to maintain mechanical integrity and porosity. Additionally, the surface of the solid MNs can be etched by electrochemical methods to form pores (Figure 4e) [81]. In this case, a hierarchical porous structure of stainless-steel MN tips could be prepared by electrochemical etching (Figure 4f). Notably, these porous metal MNs have strong mechanical strengths, but their fabrication procedures often involve sophisticated and highly expensive micromachining technologies. 

**Figure 4 ijms-24-09882-f004:**
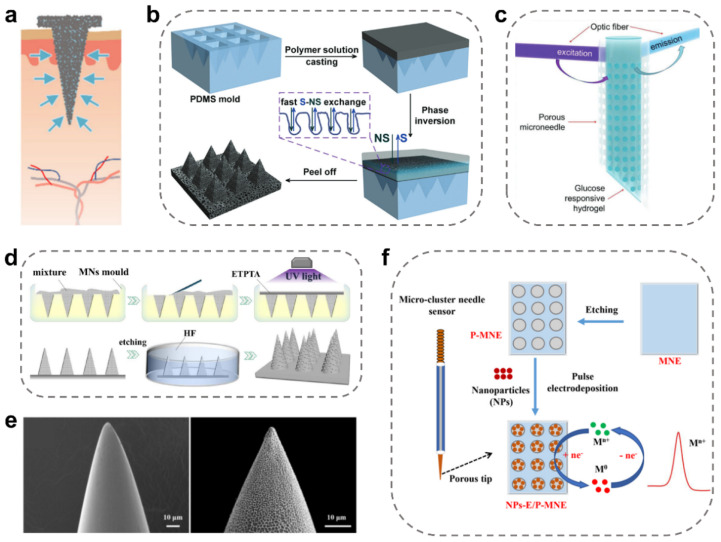
Design of porous MNs for ISF extraction. ISF extraction by porous MNs. (**a**) Schematic illustration of ISF extraction using porous MNs [12]. Reproduced with permission. Copyright 2020, Elsevier B.V. (**b**) Preparation of porous MNs via the mold method [74]. Reproduced with permission. Copyright 2012, Royal Society of Chemistry. (**c**) MN-based device with a transparent porous wall for glucose detection [53]. Reproduced with permission. Copyright 2023, John Wiley and Sons. (**d**) Schematic of the manufacture of the porous MNs by the chemical etching method [79]. Reproduced with permission. Copyright 2021, Elsevier B.V. (**e**) SEM images of the porous MNs before and after electrochemical etching [81]. (**f**) The porous fabrication of porous MNs via electrochemical etching [81]. Reproduced with permission. Copyright 2023, Elsevier B.V.

Compared with solid and hollow MNs, there are more emerging strategies to prepare porous MNs. Furthermore, porous MNs often possess high efficiency and cause less interference during ISF extraction, and they are easy to integrate with other sensors for further enhancement of the ISF extraction.

### 2.4. Coated MNs

Coated MNs are solid, hollow, and/or porous MNs coated with one or more layers of other substances on their surface, most of which serve as protective layers in extraction and detection, to enhance puncture strength, or to endow MNs with other functions [30,54,82,83,84]. Particularly, the partially coated MNs rely on the strong swelling property of the coating hydrogel for locking the MNs tips in the local tissue and preventing the MNs tips from falling off [56]. In one case, Yang et al. developed a biphasic MNs array whose tips consist of a non-swellable polystyrene (PS) core and a polystyrene-polyacrylic acid (PS-b-PAA) swellable coating (Figure 5a) [30]. Compared with the uncoated MNs, the coated ones could be secured to the skin graft site by mechanical interlocking of the expandable coated tips, resulting in a 3.5-times increase in adhesion strength. In addition, a hydrogel coating could function as a molecular filter to remove non-target substances. For example, Srinivasu Valagerahally Puttaswamy et al. coated a layer of poly (lactic-co-glycolic acid) (PLGA) hydrogel on the surface of maltose-porous MNs (Figure 5b) [85]. The porosity of PLGA was used to filter cells or other macromolecular substances in capillaries. Moreover, the developed MNs can be extended to other disease biomarkers by changing the biorecognition-coated layer.

Typically, coated MNs can quickly elicit a response from the body and extract the target from the ISF when the coating is modified by special substrates. In one example, the agarose MNs tips were modified with a catechol-immobilized coating to detect the tyrosinase (TYR, tumor markers for melanoma) enzyme [54]. The catechol could be quickly converted to benzoquinone in the presence of the TYR enzyme, which was detected perimetrically with a current signal proportional to the TYR level. Similarly, Dana Al Sulaiman et al. invented a MNs patch coated with a peptide nucleic acid-alginate hybrid hydrogel [86]. The coated-MN-based platform exhibited rapid nucleic acid collection and large sampling capacity (up to 6.5 μL within 2 min), enabling the complete detection of targeted nucleic acid biomarkers on the MN patch or in solution after light-triggered release from the material.

Compared with other types of MNs, the thinner coating of coated MNs can collect ISF rapidly in a several seconds. Moreover, the coating MNs can not only be modified on the coating, but also maintain the own characteristics of the inner MNs, such as the strong mechanical property of the solid MNs and the capillary action of the porous MNs.

## 3. Construction of MN-Integrated Sensors

In addition to the single MN-based extraction platforms, more and more efforts are being made to explore MN-integrated sensors with multiple modalities of multi-period extraction, multi-substance monitoring, and their combinations [7,87]. MNs can play a powerful role in real-time continuous monitoring after modification by other sensors (Table 2). The working principles and monitoring strategies are primarily determined by the characteristics of MN materials. In this section, we will focus on the construction of MN-integrated sensors for metabolic analysis, with highlights on the electrochemical, fluorescent, chemical chromogenic, immunodiagnostic, and molecular diagnostic MN-integrated sensors.

### 3.1. Electrochemical MN-Integrated Sensors

Electrochemical MN-integrated sensors are a kind of sensor according to the electrochemical performance of the analyte, in which the chemical quantity is converted into electric properties for sensing and monitoring [7,94,95,96]. Electrochemical biosensors well integrate electrochemistry and biotechnology and mainly consist of signal transduction, regulation, processing, and wireless transmission [97]. They were first applied for monitoring oxygen, and later they were used to detect a variety of toxic gases and displayed excellent selectivity and sensitivity [89]. Electrochemical sensors have been widely used in biomedicine and can be used to measure the concentration of important molecules (such as glucose, lactic acid, hemoglobin, etc.), monitor biological reactions (such as immune reactions, enzyme reactions, etc.), and detect the existence of pathogens [98,99,100].

Nowadays, electrochemical sensors have been integrated with MNs to detect small molecular physiological indexes in ISF. As an example, Yanxiang Cheng et al. exploited a touch-driven biosensor for collecting and monitoring glucose in ISF based on “MN transdermal-reverse iontophoresis (RI) extraction-electrochemical detection” [101]. This type of MN could enhance the RI-extraction efficiency of glucose due to the existence of microchannels. Compared to RI extraction alone, it significantly improved the glucose collecting flux by 1.6-fold. Furthermore, they also exploited a smartphone-based electrochemical device for monitoring glucose, which consists of a touch-actuated electrochemical sensor, a wireless detector, and a customized smartphone application (Figure 6a,b). This MNs-based electrochemical device has the advantages of low price and security that could be conducted by non-professional patients, and it is very suitable for home diabetic monitoring. 

In order to shorten the detection time and develop more diversified applications in other organizations, it is necessary to improve the electrocatalytic activity of electrochemical MN-integrated sensors. A recent study reported an enzyme-free platinum electrode MN-integrated sensor based on Au nanoparticle modified polydopamine nanospheres (PDA-NSs) for the detection of lactic acid in live tumors, with a detection time 7.5 times faster than that of existing clinical methods [102]. In this case, the Au nanostructure could significantly improve the electrocatalytic activity, and thus the nanohybrid microelectrode showed good sensitivity and selectivity for non-enzymatic electrochemical detection of lactic acid. This case also gives us a lot of ideas about things other than ISF detection. In another case, Huijie Li et al. reported a MN-based minimally invasive potentiometric sensor for multiplexed monitoring of Na^+^/K^+^ from cutaneous ISF [88] (Figure 6c). This potentiometric sensing system consists of a stainless steel MN with a hollow channel for preventing sensor delamination and a set of modified electrodes for multiplex monitoring (Figure 6d). A rapid response to the detection of Na^+^/K^+^ in ISF with excellent reversibility/repeatability and sufficient selectivity was observed. In addition, this MN-integrated sensor could still maintain specificity and sensitivity during inserting into the chicken skin, and its detection was not negatively affected by food, drugs, etc.

### 3.2. Fluorescent MN-Integrated Sensors

Fluorescent MN-integrated sensors typically use fluorescent dyes or fluorescent proteins as signal indicators to detect and analyze specific substances [87,103]. Their work principle is based on the change of fluorescence signals after binding or reacting with the target substance, so as to realize the detection and analysis of the target substance [104]. Compared with electrochemical MN sensors, fluorescent MN sensors are likely more stable and sensitive and could deal with the problem of electrochemical interference from body tissue. For example, Jian Zhang et al. developed a porous MN patch with a boronic hydrogel system decorated with fluorescent nanodiamonds, which could stimulate fluorescent signals during the detection of the environmental glucose concentrations (Figure 7a) [53]. Thanks to the photostable capacity of fluorescent nanodiamonds, this device displays long-term and stable visual signals in animals.

Current fluorescent sensors have restrictions on low-wavelength fluorescence and poor biosafety. Therefore, Samuel Babity et al. developed a dissolving MN-integrated sensing device for radiometric detection and imaging of reactive oxygen [105]. As compared to the MNs containing free dyes (hydrophobic molecules), the MN tattoos with PEGylated dyes significantly improve the diffusion of PEGylated dyes in the skin tissue, resulting in a uniform fluorescence distribution (Figure 7b). Furthermore, the accuracy of the fluorescent MN-integrated sensor could be improved by incorporating a fluorescence quencher within the device. For example, Xianlei Li et al. described a fluorescence-amplified origami MN (FAO MN) device with an internal network structure containing proton-driven fluorophore, quencher-containing DNA pairs, and glucose oxidase molecules for orderly monitoring glucose from ISF [90]. The FAOM sensor could detect glucose in MN structures and convert it to a proton signal via oxidase’s catalysis. Ultimately, the fluorescent molecules and quenchers were separated by the proton-driven mechanical reconstitution, magnifying the glucose-mediated fluorescence signal. Compared with the commercial blood biochemical analyzer, FAO MNs reached 98.70 ± 4.77% accuracy and absolutely satisfied the requirements of continuous glucose monitoring. Fluorescence can also be enhanced by optimizing the structure of MNs [57]. For instance, Kexin Yi et al. developed a novel inverse opal MNs array [91] (Figure 7c), which displayed a porous tip with a particularly arranged structure and special photonic band gap (PBG) characters (Figure 7d). The MNs patch with an inverse opal structure enables enhanced targeted bio-probe fluorescence signal intensity and improves detection sensitivity. These results indicate the great potential value of these MN-integrated fluorescence sensors in non-invasive clinical diagnosis.

### 3.3. Chemical Chromogenic MN-Integrated Sensors

Chemical chromogenic MN-integrated sensors are a kind of chemical sensors that realize the detection and quantitative analysis of the target molecule based on the changeable color after the specific and chemical reaction between the target detector and microneedle sensor [51,106,107]. The developed device usually consists of an indicator, coordination groups, and recognition elements and could conduct the visual diagnosis using the red, green, and blue (RGB) analysis system. These early chromogenic sensors were used to detect sweat on the skin surface for assessing the body’s metabolism. For example, Jingyu Xiao et al. developed a microfluidic colorimetric MN device [106]. The sweat on the skin surface was routed through the microchannels to the microchambers containing glucose oxidase-peroxidase-o-dianisidine reagents for detecting the glucose in sweat. In recent years, metabolites in sweat have also been used to assess the physiological state of the body. However, it may not be applicable to sweatless people, and it is not as stable as skin ISF test results because of the influence of sweat flow.

At present, the combination of chemical colorimetric sensors and MNs with in situ extraction functions has been widely used in biomedical safety fields. For example, a MeHA MN was integrated with a sensing-reagent-decorated test paper (TP) for multiplexed colorimetric detection of glucose, lactate, cholesterol, and pH (Figure 8a) [92]. This TP-MeHA MN bio-device generated sudden color changes in a concentration-dependent way and allowed diagnosis by the naked eye or RGB quantitative analysis with a colorimeter. Similarly, based on the specific reactions for colorimetric detection, Mahmood Razzaghi et al. presented a PEGDA-MN integrated with a multiplexed sensing device, providing a color change in a concentration-dependent way [58]. In this study, pH levels about 7.0–10.0 could be recorded calorimetrically using this MNs device (Figure 8b), while glucose concentrations about 0–12 mM could be monitored by MNs (Figure 8c). Such chemical colorimetric MN-integrated sensors facilitate a convenient and self-administrable profiling of metabolites and shall be instrumental for home-based long-term monitoring and management of metabolic diseases.

### 3.4. Immunodiagnostic MN-Integrated Sensors

Immunodiagnostic MN-integrated sensors are committed to detecting the humoral or cellular immune response based on the extraction action of MNs and the analytic functions of immunological techniques [57,108,109]. They could determine whether there is an infection with a pathogenic microorganism, a tumor, or an autoimmune disease. Generally, the antigen (Ag) or antibody (Ab) are used to detect the target Ag/Ab in the sample via a specific Ag-Ab binding reaction. The detection results are usually amplified by fluorescence, chemiluminescence, chemical color development, etc. [51,75,96]. Nowadays, MNs are mainly applied to the extraction/transfer of ISF clinical specimens, which could be transformed from MNs into an effective immunodiagnostic sensor. The immunodiagnostic MN-integrated sensors can realize in situ immunodetection in local tissues [110].

Today, a series of MNs based on Ag-Ab-specific binding have been developed for distinguishing specific protein markers from cutaneous ISF. These MNs are stripped of their skin and incubated with enzyme-labeled antibodies for a few minutes. The targeted protein markers are subsequently observed and quantified by fluorescent microscopy or enzyme-substrate coloration. For example, Zhang et al. proposed novel immunodiagnostic MNs integrated with photonic crystal (PhC) barcodes that enriched specific Ab onto their probe-modified PhC balls [57]. The captured Ag could be easily captured and discerned by observing the reflective colors of PhC barcodes, and these corresponding fluorescence intensities indicate the relative biomarker amounts. Such MNs enabled multiplex specific detection of ISF biomarkers (IL-1β, TNF-α, and IL-6) in a mouse model of sepsis (Figure 9a). This work indicated that MNs with multifarious Ag/Ab integration are promising for disease marker monitoring and screening.

Cell populations in local tissues cannot often be monitored by traditional blood draws. The concentration of antigen-specific T cell populations in the systemic circulation (blood) is extremely low and of strong clinical significance, especially at the time of vaccination or infection. Therefore, in situ T-cell detection by immunodiagnostic MNs is of great significance for monitoring the local immune response. Anasuya Mandal et al. designed a microneedle array loaded with adjuvant and specific Ag to draw in antigen-presenting cells (APC) in skin for subsequent cell phenotype and functional analysis without changing the immune status of local tissues [93]. This method obtains biopsy information from skin or other mucosal tissue by enriching APC in a minimally invasive manner (Figure 6b). At present, increasing studies have been conducted to improve the sensitivity of immune responses by reforming MN surfaces and magnifying the surface area inside MNs.

### 3.5. Molecular Diagnostic MN-Integrated Sensors

Molecular diagnostic MN-integrated sensors provide a novel method for in situ detection, analysis, and diagnosis of genetic materials in tissues by integrating MNs with molecular biology technologies [32,33,111]. They can detect abnormal changes in molecules such as DNA, RNA, etc., which offer evidence for early diagnosis, prevention, and treatment of diseases. 

In situ detection of microRNA (miRNA) in tissues has become a new approach for clinical health monitoring, or in situ or local molecular diagnosis [112]. In one case, Qiqi Yang et al. developed a MeHA-MNs patch containing a smart DNA hydrogel system to achieve rapid enrichment of miRNAs [56]. When the MeHA/DNA-MNs patch enriched a certain amount of miRNA, a cascade of DNA displacement reactions would be triggered, and fluorescence would be generated for detection. Specifically, the signal amplification response with cascade endpoint-mediated DNA displacement allowed miRNA monitoring with a sensitivity as low as 241.56 pM. In another case, Dana Al Sulaiman et al. developed MN arrays coated with a sodium alginate (SA)-peptide nucleic acid (PNA) hydrogel for simultaneous in situ detection of the specific miRNA markers from cutaneous ISF (Figure 10) [86]. Compared with standard oligonucleotides, PNAs had a stronger affinity and sequence specificity during hybridization to complementary DNA or RNA. As shown in Figure 10b, when the MN patch was conducted to extract a solution with DNA or RNA, the targeted sequences could be hybridized by the PNA probe, and any non-specific molecules could be washed from the hydrogel matrix. Such MNs have fast sample kinetics and a large capacity (3.25 μL/min), enabling the detection and quantification of targeted nucleic acids at once in the MN itself or in a light-triggered hydrogel solution. In addition, Bin Yang et al. invented an on-line wearable MN patch with CRISPR-Cas9-activated graphene bio-interfaces for in vivo continuous monitoring of DNA [111]. In their research, the wearable device enabled real-time detection of the Epstein-Barr virus and kidney transplantation cell-free DNA and had excellent stability for 10 days in vivo. This molecular diagnostic MN-integrated sensor holds great potential for continuous and long-term in vivo monitoring of cell-free DNA and could possibly be used for early disease screening and postoperative observation. 

## 4. Conclusions and Perspectives

MN-integrated sensors have been recognized as powerful platforms for skin ISF extraction and detection applications [113,114,115]. In this manuscript, we review the progress of the MN-integrated sensors for ISF extraction and metabolic analysis in the past decade [116,117,118,119]. Following the introduction of the structure design of MNs for ISF extraction, we outlined five typical types of MN-integrated sensors, including electrochemical, fluorescent, chemical chromogenic, immunodiagnostic, and molecular diagnostic-MN-integrated sensors. Compared with the traditional sensors, MN-integrated sensors have the following advantages [120,121,122]. First, no special equipment is needed. In addition, bioavailability and detection efficiency are improved [123]. Moreover, portable and precisely customized platforms could be developed. Compared with traditional fabrication methods, 3D printing presents a significant advance for fabricating MNs in a reproducible fashion with high resolution and quality based on the programmed procedures. Moreover, 3D printing allows the fast modification of the key properties that play an important role in the performance of MNs, such as the needle height, tip radius, base diameter, needle geometry, needle thickness, and needle density. With its huge potential, 3D printing technology has become a promising new tool to help create novel designs, improve efficacy, and increase the functionality of MNs. It is worth mentioning that less painful injections are involved because the tiny needles penetrate the skin or mucous membranes to reduce or eliminate the sting. These unique features render MNs a portable detector for effective extraction and detection of the biomarkers in ISF.

The integration of advanced MNs and multiple sensors presents a promising ISF-detection platform for clinical transformations [51,92]. However, significant developments on several issues are still anticipated. For example, the biocompatibility and long-term stability between MN-based sensors and skin tissues remain to be improved in future practical applications. Additionally, current fabrication methods often involve multi-step procedures, which might be difficult to replicate for different manufacturers. Another issue is to further improve the sensitivity and selectiveness of the MN-based sensors. Thus, new designs of both MNs and biosensors with greater efficiency and accuracy should be exploited instead of relying on the existing forms [12,13,123,124,125,126,127]. Future research directions should include the development of (i) more effective MNs for quick ISF collection, (ii) more rapid, simple but intelligent sensors for metabolic analysis, (iii) advanced technologies to fabricate MN-integrated sensors, (iv) MN-based new therapeutic and diagnostic strategies, and so on. We expect this review will inspire more breakthroughs in MN-integrated sensors, finally leading to more advanced platforms for biomarker detection and monitoring applications.

## Figures and Tables

**Figure 1 ijms-24-09882-f001:**
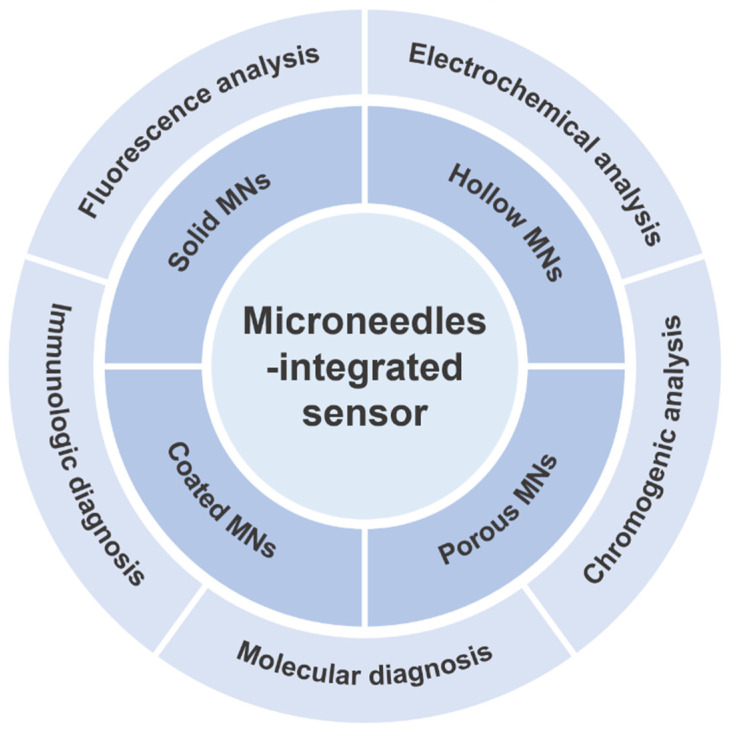
Schematic illustration of typical microneedle (MN)-integrated sensors for extraction of skin interstitial fluids and metabolic analysis.

**Figure 2 ijms-24-09882-f002:**
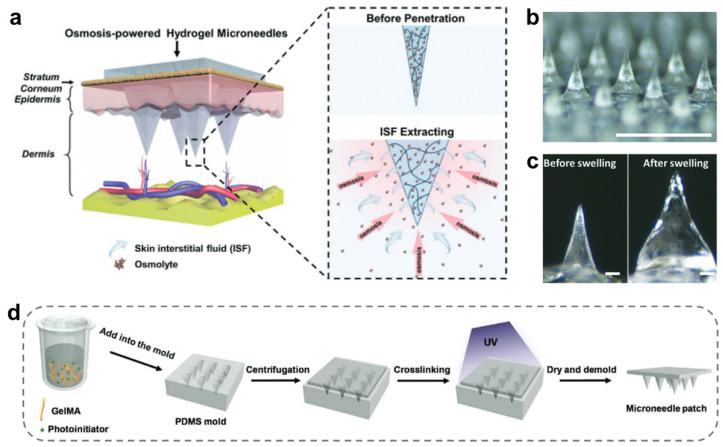
Design of solid MNs for ISF extraction. (**a**) Schematic illustration of the skin ISF extraction using osmolyte-containing hydrogel MNs [47]. Reproduced with permission. Copyright 2020, John Wiley and Sons. (**b**) Optical image of solid hydrogel MN patches (scale bar = 1 mm) [37]. (**c**) Optical microscopic images showing the hydrogel MNs before and after swelling (scale bar = 100 μm) [37]. Copyright 2021, John Wiley and Sons. (**d**) Schematic illustration of the preparation process of photopolymerized hydrogel solid MNs [48]. Reproduced with permission. Copyright 2020, John Wiley and Sons.

**Figure 3 ijms-24-09882-f003:**
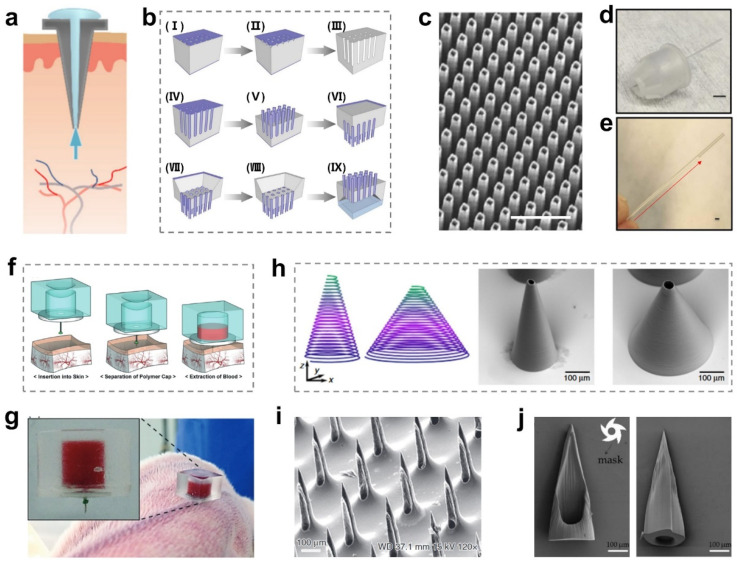
Design of hollow MNs for ISF extraction. (**a**) Schematic illustration of ISF extraction using hollow MNs [12]. Reproduced with permission. Copyright 2020, Elsevier B.V. (**b**) Schematic illustration of the fabrication process of hollow MNs via etching methods [64]. (I) patterning of a lattice of square holes on the front-side silicon-dioxide layer, by standard lithography and BHF etching; (II) formation of pyramidal notches on the front-side silicon surface, by KOH etching; (III) formation of an array of regular and deep macropores, by BIEE; (IV) formation of silicon-dioxide pipes embedded into the silicon substrate, by wet thermal oxidation; (V) production of an array of microneedles protruding from the front-side silicon surface, by KOH etching; (VI) patterning of square windows on the back-side silicon-dioxide layer, by standard lithography and BHF etching; (VII) reservoir fabrication on the back-side of the silicon die, by TMAH etching; (VIII) removal of the silicondioxide cap at the bottom of microneedles, by BHF etching; (IX) bonding of a plastic cover on the back-side of the silicon die. Light grey is silicon and dark grey (violet on-line) is silicon dioxide. Reproduced with permission. Copyright 2001 Royal Society of Chemistry. (**c**) SEM images of hollow silicon-dioxide MNs (scale bar = 50 μm) [50]. Reproduced with permission. Copyright 2015, Elsevier B.V. (**d**) Single hollow steel MNs with a glass capillary collection tube (scale bar = 5 mm) [6]. (**e**) The ISF in the capillary tube was collected by hollow MNs (scale bar = 1 mm) [6]. (**d**,**e**) reproduced with permission. Copyright 2018, Springer Nature. (**f**) Schematic illustration of the blood sampling process using hollow MN tips and a chamber [65]. (**g**) The blood sample was extracted into the chamber with hollow MN tips [65]. (**f**,**g**) reproduced with permission. Copyright 2001, Royal Society of Chemistry. (**h**) CAD design and SEM images showing thin and thick truncated cone-shaped hollow MNs (scale bar = 100 μm) [66]. Reproduced with permission. Copyright 2019, Springer Nature. (**i**) The SEM image of hollow MNs after wet etching and before plasma etching (scale bar = 100 μm) [67]. Reproduced with permission. Copyright 2019, Springer Nature. (**j**) SEM microstructures of hollow MN tips after etching (scale bar = 100 μm) [68].

**Figure 5 ijms-24-09882-f005:**
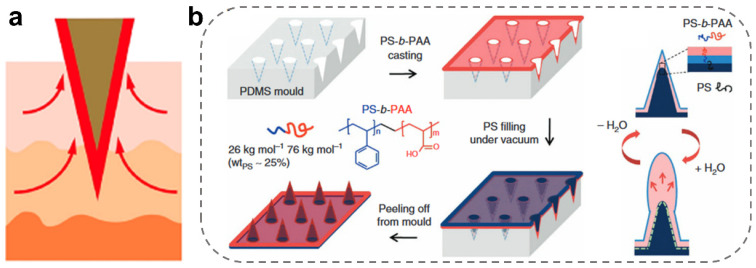
Design of coated MNs for ISF extraction. (**a**) Schematic illustration of ISF extraction using coated MNs [30]. Reproduced with permission. Copyright 2013, Springer Nature. (**b**) Preparation of the coated MN array with swellable tips via the mold method [85]. Reproduced with permission. Copyright 2020, American Chemical Society.

**Figure 6 ijms-24-09882-f006:**
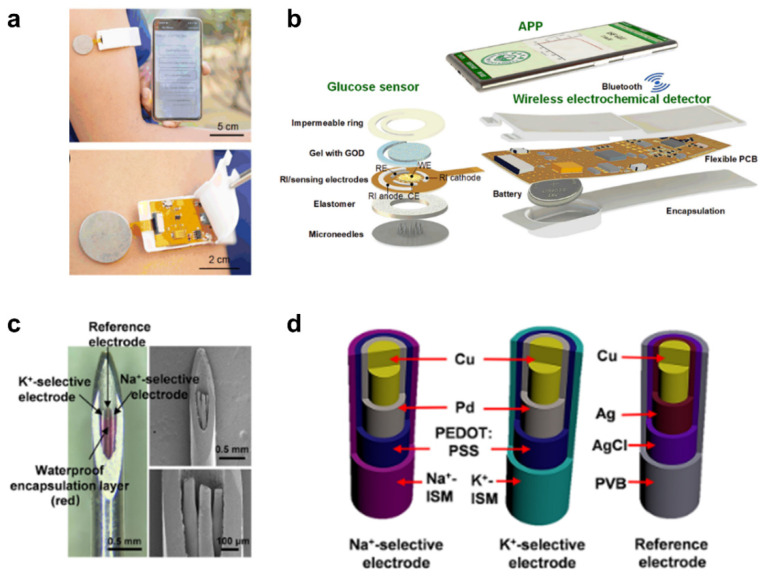
Construction of electrochemical MN-integrated sensors. (**a**) Photograph of the smartphone-based glucose electrochemical MN-integrated sensor [101]. (**b**) Schematic diagram showing its compositions, including the touch-actuated glucose sensor integrated with MNs and RI, the wireless electrochemical detector, and the Android-based smartphone APP [101]. Reproduced with permission. Copyright 2022, Elsevier B.V. (**c**) Optical and scanning electron microscope images of an MN-integrated potentiometric sensor [88]. (**d**) Schematic illustration of the sodium/potassium ion-selective electrodes and the reference electrode with multi-layer modifications inside the hollow MNs [88]. Reproduced with permission. Copyright 2021, American Chemical Society.

**Figure 7 ijms-24-09882-f007:**
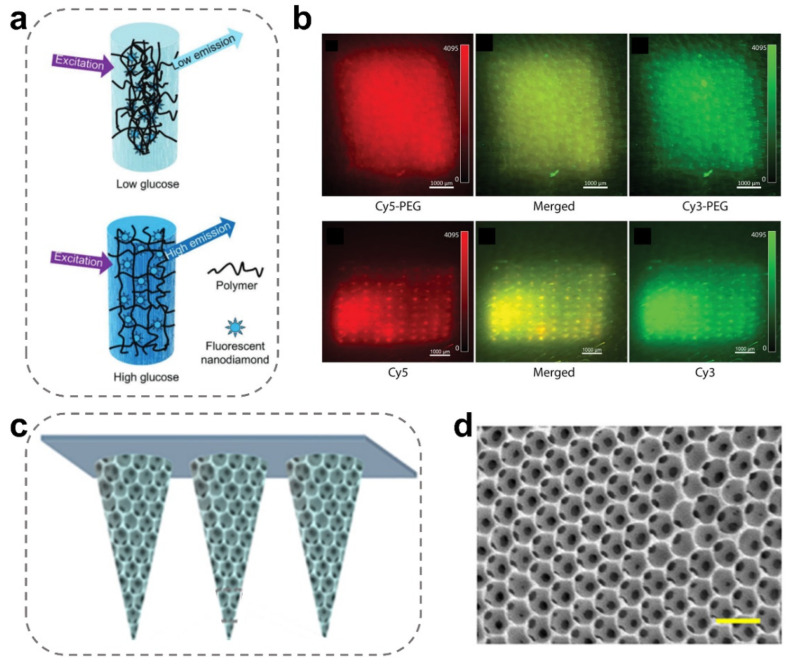
Construction of fluorescent MN-integrated sensors. (**a**) Schematic illustration of the fluorescent MN-integrated device for quantitatively monitoring blood glucose [53]. Reproduced with permission. Copyright 2023, John Wiley and Sons. (**b**) The inverse opal MN array with a fluorescence-enhanced effect [105]. Reproduced with permission. Copyright 2021, John Wiley and Sons. (**c**,**d**) Scheme and SEM images of the micro-porous structure of inverse opal MNs (scale bar = 300 μm) [91]. Reproduced with permission. Copyright 2022, Elsevier B.V.

**Figure 8 ijms-24-09882-f008:**
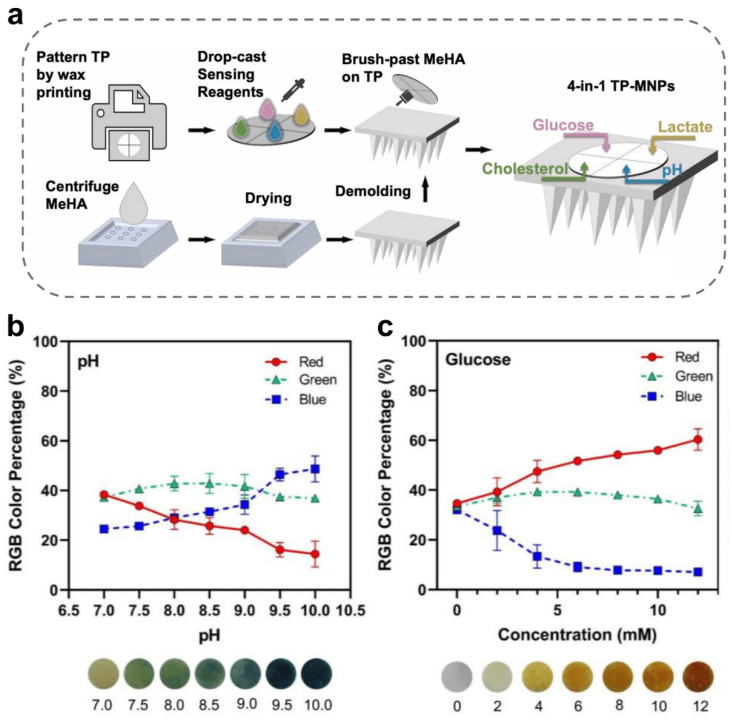
Construction of chemically chromogenic MN-integrated sensors. (**a**) Schematic illustration showing the fabrication and application process of test-paper-incorporated microneedle patches (TP-MeHA MNs) [92]. Reproduced with permission. Copyright 2022, Elsevier B.V. (**b**,**c**) RGB color percentage changes for pH (**b**) and glucose sensing (**c**) [58].

**Figure 9 ijms-24-09882-f009:**
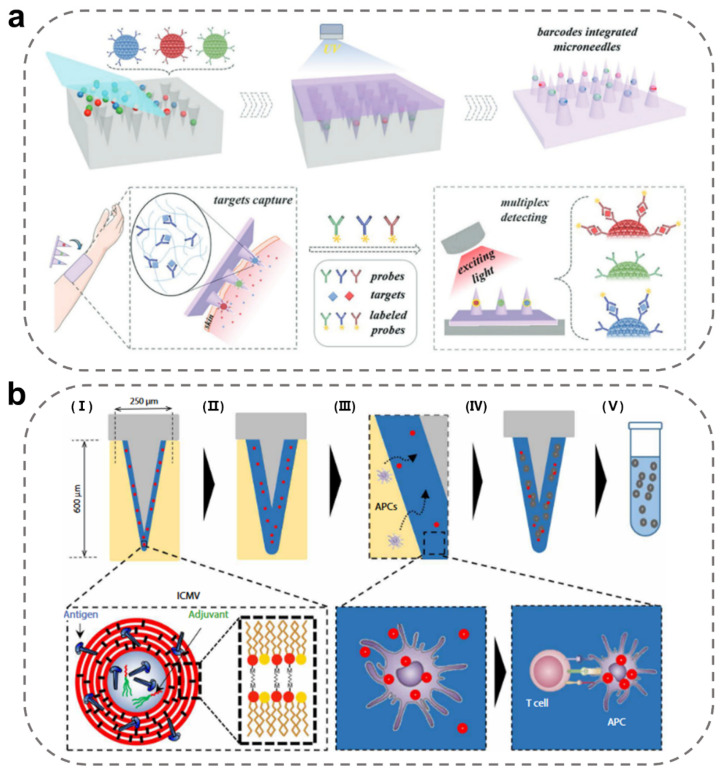
Construction of immunodiagnostic MN-integrated sensors. (**a**) Schematic illustration of immunodiagnostic MN fabrication and application in ISF detection [57]. Reproduced with permission. Copyright 2019, John Wiley and Sons. (**b**) Scheme of the structure and mechanism of enriching-APC MNs [93]. (I) MN with dried hydrogen coating, applied to skin. (II) Swelling of hydrogel layer with interstitial fluid. (III) Immune cell infiltration into hydrogen layer. (IV) Removal of MN array from skin. (V) Un–cross-link hydrogel layer and release cells and interstitial fluid for analysis. Reproduced with permission. Copyright 2018, The American Association for the Advancement of Science.

**Figure 10 ijms-24-09882-f010:**
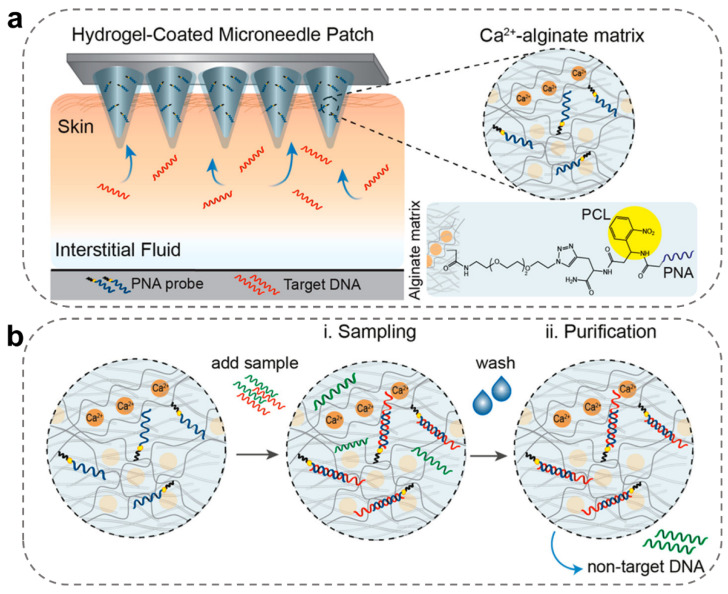
Construction of molecular diagnostic MN-integrated sensors. (**a**) Schematic illustration of MNs functionalized with SA-PCL-PNA coatings for detecting target DNA in skin ISF [86]. (**b**) The generic protocol for target DNA sampling and purification to remove nontarget sequences. Circles represent a magnification of the SA hydrogel coating on the MN patches [86]. Reproduced with permission. Copyright 2019, American Chemical Society.

**Table 1 ijms-24-09882-t001:** Representative MNs with various structures for ISF extraction.

Structures	Materials	Extracts	Fabrication Methods	Refs.
Solid MNs	Polyvinyl alcohol and chitosan	Glucose, chlorine, lactate	Micro-molding method	[46]
	Methacrylated hyaluronic acid	Glucose and cholesterol	Micro-molding method	[11]
	Maltose and methacrylated hyaluronic acid	Glucose	Micro-molding method	[47]
	Gelatin methacryloyl	Glucose and vancomycin	Micro-molding method	[48]
Hollow MNs	Polydimethylsiloxane (PDMS)	Glucose and lactic acid	Three-dimensional (3D) printer and micro stereolithography (PμSL) technology	[49]
	Silicon-dioxide	Glucose	Standard lithography, buffered hydrofluoric acid (HF) and potassium hydroxide (KOH) etching et al.	[50]
	Stainless steel	Exosomes	Photoetching	[6]
	Polymerized SU-8 photoresist	Plasmodium falciparum histidine-rich protein 2	Photoetching	[51]
Porous MNs	Stainless steel	Glucose	Sintering at 1100 °C and subsequent electropolishing	[52]
	Dopamine and hyaluronic acid	Glucose	Micro-molding method	[36]
	Acrylic resin	Glucose	Salt leaching method	[53]
Coated MNs	Poly lactic-co-glycolic acid (coated), maltose (solid)	Cystatin C	Drawing and wrapping method	[30]
	Catechol(coated), photocurable acrylate (hollow)	Tyrosinase enzyme	Coating method	[54]

**Table 2 ijms-24-09882-t002:** Representative MN-integrated sensors with different working principles for metabolic analysis.

Classification	Materials	Extracts	Examples
Electrochemical MN-integrated sensors	Stainless-steel (hollow)	Na^+^ and K^+^	Hollow MN-based potentiometric sensor consisting of a sodium/potassium ion-selective electrode and an Ag/AgCl reference electrode. Ref. [88].
	Hyaluronic acid	pH	Solid MNs made up of dopamine (DA) conjugated HA, and poly(3,4-ethylenedioxythiophene): polystyrene sulfonate (PEDOT: PSS) to increase conductivity. Ref. [89].
Fluorescent MN-integrated sensors	Maltose and methacrylated hyaluronic acid	Glucose	The porous MNs were integrated with a fluorescent nanodiamond boronic hydrogel system. Ref. [90].
	Acrylic resin (porous)	Glucose	Surface functionalization using the fluorescent nanodiamond embedded in the boronic polymer hydrogels. Ref. [53].
	Ethoxylated trimethylolpropane triacrylate (porous)	Lipopolysaccharide	The MNs had an inverse opal structure with fluorescence enhanced signal. Ref. [91].
Chemical colorimetric MN-integrated sensors	Methacrylated hyaluronic acid and hyaluronic acid (solid)	Glucose, lactate, cholesterol, and pH	The MeHA MNs was incorporated with multiplexed colorimetric and sensing-reagent-decorated test paper. Ref. [92].
Immunodiagnostic MN-integrated sensors	poly(ethylene glycol) diacrylate (solid)	TNF-α, IL-1β, and IL-6	PEGDA MNs containing photonic crystal barcodes connecting specific antibody. Ref. [57].
	Poly-l-lactide (solid), alginate (coated)	Memory T cells	The immune adjuvants and specific antigens nanocapsules were embedded in the MN coating. Ref. [93].
Molecular diagnostic MN-integrated sensors	Methacrylate hyaluronic acid (solid)	miRNA	The MeHA MNs were equipped with the DNA displacement signal amplification system. Ref. [56].
	Alginate, Poly-L-Lactide (coated)	miRNA	The surface of Poly-L-Lactide MNs were coated with an alginate-peptide nucleic acid hybrid system for sequence-specific sampling. Ref. [86].

## Data Availability

Not applicable.

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
