# Peer review of "Microneedle-Integrated Sensors for Extraction of Skin Interstitial Fluid and Metabolic Analysis"

_ijms, 2023, doi:10.3390/ijms24129882_

Round 1
Reviewer 1 Report
The paper is a review on microneedle-integrated sensors for extraction of skin interstitial fluid and analysis. The paper is well written and I suggest only minor revisions. In particular, for the sake of completeness, I suggest the authors to include and cite the very recent paper published by Miranda, B., Battisti, M., De Martino, S., Nocerino, V., Dardano, P., De Stefano, L. and Cangiano, G. (2023), Hollow Microneedle-based Plasmonic Sensor for on Patch Detection of Molecules in Dermal Interstitial Fluid. Adv. Mater. Technol. 2300037. https://doi.org/10.1002/admt.202300037. The Fig. 3 panel j is not commented in the caption of the figure; probably panel g is twice cited, the last should be j, but please check it.
The English style and form are good enough
Author Response
Thank the reviewers for the positive comment on our manuscript. According to the reviewer’s suggestions, the manuscript has been carefully revised.
Response 1: As the reviewer suggested, we have carefully read the recommended references, which are helpful and have been added to the revised manuscript. (Revised Ref.123 on page 24)
Response 2:As the reviewer suggested, we have carefully checked the sequence numbers, and revised the repetitive Fig. 3g to Fig. 3j in the revised manuscript. (Revised in paragraph 1 on page 6)
Reviewer 2 Report
The manuscript summarizes the information on the microneedle-integrated sensors and their application for the extraction from interstitial fluid and appropriate metabolics analysis. The manuscript can be published in International Journal of Molecular Sciences after minor changes:
1. Abstract partially duplicates the Introduction. Please rewrite the Introduction.
2. In Figure 1: “analyze” should be changed with “analysis”.
3. Page 11: not “Figure 5a,b” but “Figure 6 a, b”.
4. There are some references ([65], [90], [106], [102]) which cannot be attributed to the metabolite detection in ISF. Validate their necessity in the manuscript and add some text in the Introduction section like that: "Some examples of microneedle sensors developed for other biological samples investigations are also presented in the manuscript.”
5. Page 14:” while the detecting concentrations” – word “glucose” is missing.
6. Page 3: “ without painless sense” –confusing phrase. Rewrite it.
Author Response
Thank the reviewers for the positive comment on our manuscript. According to the reviewer’s suggestions, the manuscript has been carefully revised.
Response 1:As the reviewer suggested, we have revised the duplicative description of paragraph 3 in the Introduction.
In this review, we will systematically describe the construction and related operating principles of MNs integrated sensors for ISF extraction and metabolic analysis (Figure 1). We begin with the structural design of MNs for collecting and extracting ISF. Then we elaborate on the construction of MN-integrated sensors for metabolic analysis with other detection methods. Finally, we discuss the potential for future development of MN-based platforms for ISF extraction and sensing. We hope this review could lay a foundation for the research of advanced MN-integrated sensors for analysis of biomarkers in skin ISF and facilitate their translation for future clinical applications.
(Revised in paragraph 2 on page 2)
Response 2:As the reviewer suggested, we have revised the “analyze” to “analysis” in Figure 1 in the revised manuscript.
(Revised in Figure 1 on page 2)
Response 3:As the reviewer suggested, we have revised the “Figure 5a,b” into “Figure 6 a,b” in Page 11 in the revised manuscript.
(Revised in paragraph 2 on page 9)
Response 4:As the reviewer suggested, we have carefully checked the relative references, and supplemented necessary description in the Introduction in revised manuscript.
In Ref.65, Cheng Guo Li et al. introduced a blood extraction device, which was fabricated by connecting a hollow MNs and a pre-vacuum PDMS actuator (Figure 3f). Although the device is not attributed to the metabolite detection in ISF, it presents a good example to integrate negative pressure devices with MNs to enhance the MN extraction efficiency. The blood was still the gold standard sample medium. If we could extract blood directly from capillaries using MNs, which thereby facilitate the detection of abundant useful biological information for the diagnosis of various diseases.
In Ref.90, we have carefully checked the descriptions about this example and corrected the unintended mistakes in the revised manuscript. Xianlei Li et al. described a fluorescence-amplified origami MN (FAO MN) device with internal network structure containing proton-driven fluorophore, quencher-containing DNA pairs and glucose oxidase molecules for orderly monitoring blood glucose. The description of “for orderly monitoring blood glucose” has been revised to “for orderly monitoring glucose from ISF”.
The Ref.102 reported an enzyme-free platinum electrode MN-integrated sensor based on Au nanoparticle modified polydopamine nanospheres (PDA-NSs) for the detection of lactic acid in live tumors, with a detection time 7.5 times faster than that of existing clinical methods. This study provides a good example to shorten the detection time and improve the electrocatalytic activity of electrochemical MN-integrated sensors.
In Ref.106, Jingyu Xiao et al. developed a microfluidic colorimetric MN device, which was described to show a paradigm of chemical chromogenic MN-integrated sensors. This MN device allowed the sweat on skin surface routed through the microchannels to the microchambers containing glucose oxidase-peroxidase-o-dianisidine reagents for the detection and quantitative analysis of the glucose in sweat.
Although the above two examples of microneedle sensors (Ref. 102 and 106) were developed for other biological samples instead of ISF, the design concepts and working principles behind them could provide constructive inspirations for the development of more advanced MN-integrated sensors for extraction and metabolite detection of ISF.
(Revised in paragraph 1 on page 2, paragraph 1 on page 5, paragraph 3 on page 10, paragraph 3 on page 9, and paragraph 3 on page 11)
Response 5:As the reviewer suggested, we have supplemented the word “glucose” in the revised manuscript.
(Revised in paragraph 1 on page 12)
Response 6:As the reviewer suggested, we have rewritten the confusing sentence in the revised manuscript.
The sentence “MNs are small needles and have a short penetration depth, providing a minimally invasive method without painless sense” has been corrected as “MNs consist of a series of micron-scale needles with shallow penetration depth, providing a minimally invasive method to puncture the skin in a painless or slightly painful manner”.
(Revised in paragraph 4 on page 2)
Reviewer 3 Report
The review article " Microneedle-integrated sensors for extraction of skin interstitial fluid and metabolic analysis" by Yang et al. discusses developments of transdermal microneedle-integrated sensors for collecting interstitial fluid and for detecting various biomarkers. The authors classified microneedles for the aforementioned application and then discussed various types of sensing techniques used in microneedle-integrated sensors. The manuscript's topic is appealing because microneedles are revolutionary devices for detecting body fluids. The review work seems comprehensive. However, the modifications below are suggested.
1. Consistent names should be used in Figure 1. "Hollow MN" could be changed to "Hollow MNs" to be consistent with other mentioned types of microneedles.
2. The complete form of all abbreviations should be mentioned in the manuscript. Like HPLC, SEM, and ELISA, etc.
3. The abbreviated form of polydimethylsiloxane (PDMS) could be used after its first mention in the manuscript (Table 1). Also, this abbreviated form of PDMS can be used instead of a complete form on page 5.
4. The abbreviated form should be followed after the first time mentioning the complete form. On page 3 (Table 1), for example, the first mention of "3D" does not have the complete form, while the second mention on page 6, does.
5. The authors should check if PLGA is the correct abbreviation for poly (sodium lactate) on page 8 (last line). It does not seem correct.
6. Authors can discuss 3D printing as a revolutionary fabricating method for microneedle patches because it has many advantages over traditional fabrication methods, such as the ability to fabricate customized, more sophisticated microneedles and microneedle-integrated devices in a single step.
English is good.
Author Response
Thank the reviewers for the positive comment on our manuscript. According to the reviewer’s suggestions, the manuscript has been carefully revised.
Response 1:As the reviewer suggested, we have revised the “Hollow MN” to “Hollow MNs” in Figure 1.
(Revised in Figure 1 on page 2)
Response 2:As the reviewer suggested, we have supplemented the complete forms of all these abbreviations in our revised manuscript, such as high performance liquid chromatography (HPLC), scanning electron microscopy (SEM), enzyme linked immunosorbent assay (ELISA), etc.
(Revised in paragraph 2 on page 3)
Response 3:As the reviewer suggested, we have provided the abbreviated form of polydimethylsiloxane (PDMS) in the Table 1 and used its abbreviated form of PDMS instead of a complete form on page 5 in the revised manuscript.
(Revised in table 1 on page 3 and paragraph 2 on page 4)
Response 4:As the reviewer suggested, we have supplemented the abbreviations of three-dimensional (3D) in table 1 on page 3, and deleted the complete forms of 3D on page 6 in the revised manuscript.
(Revised in table 1 on page 3 and paragraph 2 on page 5)
Response 5:As the reviewer suggested, we have verified the abbreviation for PLGA on page 8, and revised the “poly (sodium lactate)” to “Poly (lactic-co-glycolic acid)” in the revised manuscript.
(Revised in paragraph 3 on page 7)
Response 6:As the reviewer suggested, we have further discussed the advantages of 3D printing for fabricating microneedle patches in the revised manuscript.
Compared with the traditional fabrication methods, 3D printing presents a significant advance for fabricating MNs in a reproducible fashion with high resolution and quality based on the programmed procedures. Moreover, 3D printing allows the fast modification of the key properties that play an important role on the performance of MNs, such as the needle height, tip-radius, base diameter, needle geometry, needle thickness, and needle density. With its huge potential, 3D printing technology has become a new promising tool to help create novel designs, improve efficacy and increase the functionality of MNs.
(Revised in paragraph 1 on page 15)